# Monitoring of *Paenibacillus larvae* in Lower Austria through DNA-Based Detection without De-Sporulation: 2018 to 2022

**DOI:** 10.3390/vetsci10030213

**Published:** 2023-03-10

**Authors:** Elfriede Wilhelm, Irina Korschineck, Michael Sigmund, Peter Paulsen, Friederike Hilbert, Wigbert Rossmanith

**Affiliations:** 1NOE-Animal Health Service, Schillerring 13, 3130 Herzogenburg, Austria; 2Ingenetix GmbH, Arsenalstraße 11, 1030 Vienna, Austria; 3Clinic of Ruminants, Department of Farm Animals and Veterinary Public Health, University of Veterinary Medicine, Veterinaerplatz 1, 1210 Vienna, Austria; 4Institute of Food Safety, University of Veterinary Medicine, Veterinaerplatz 1, 1210 Vienna, Austria

**Keywords:** American foulbrood, *Paenibacillus larvae*, spore, RT-PCR

## Abstract

**Simple Summary:**

American foulbrood is a deadly disease specific for honey bees. The infectious agent, *Paenibacillus larvae*, is a spore-forming bacterium. Symptoms of the disease are often seen at a very late stage. Early detection, before the disease has done any harm, is useful in controlling the burden of the disease. This is accomplished in monitoring and surveillance programs often organized by local authorities. Detection of this bacterium is time-consuming as it involves a de-sporulation process. Here, we describe a five-year voluntary monitoring program in a western district of Lower Austria based on traditional pathogen detection and direct DNA-based detection without the need for de-sporulation.

**Abstract:**

American foulbrood is caused by the spore-forming *Paenibacillus larvae*. Although the disease effects honey bee larvae, it threatens the entire colony. Clinical signs of the disease are seen at a very late stage of the disease and bee colonies are often beyond saving. Therefore, through active monitoring based on screening, an infection can be detected early and bee colonies can be protected with hygiene measures. As a result, the pressure to spread in an area remains low. The cultural and molecular biological detection of *P. larvae* is usually preceded by spore germination before detection. In this study, we compared the results of two methods, the culture detection and RT-PCR detection of DNA directly isolated from spores. Samples of honey and cells with honey surrounding the brood were used in a five-year voluntary monitoring program in a western part of Lower Austria. DNA-extraction from spores to speed up detection involved one chemical and two enzymatic steps before mechanical bashing-beat separation and additional lysis. The results are comparable to culture-based methods, but with a large time advantage. Within the voluntary monitoring program, the proportion of bee colonies without the detection of *P. larvae* was high (2018: 91.9%, 2019: 72.09%, 2020: 74.6%, 2021: 81.35%, 2022: 84.5%), and in most *P. larvae*-positive bee colonies, only a very low spore content was detected. Nevertheless, two bee colonies in one apiary with clinical signs of disease had to be eradicated.

## 1. Introduction

The causative agent of American foulbrood (AFB) in honey bees is the spore-forming bacterium *Paenibacillus larvae* (*P. larvae*). In the European Union and many other countries, infection caused by *P. larvae* is a notifiable animal disease and causes significant losses in honey bee colonies. *P. larvae* spores can survive in the environment as infectious agents for many years. Only young honey bee larvae—up to 36 h after hatching—are susceptible to infection by *P. larvae* [1,2]. As a mechanism of natural resistance, adult bees produce certain inhibitors in the gut that can prevent the growth of *P. larvae* [3].

The dispersal of *P. larvae* occurs mainly through the transmission of spores by adult bees particularly by flying back and forth between neighboring apiaries and hives, and by predatory bees attacking weakened colonies [4,5,6]. Spores can be found in the brood, honey, beeswax, pollen and on the walls of the infected hives, and germination of spores can be successful even after 35 years [6,7]. Within hives, spores of *P. larvae* are transmitted through the feeding of larvae by adult bees. Certain habitual techniques used by beekeepers can promote spore transmission, e.g., the replacement of brood combs or entire bee colonies, or beekeeping in confined spaces [8]. *P. larvae* spores can also be found in beehive debris and honey from infected bee colonies without noticeable clinical symptoms [9]. 

Classical symptoms of AFB in diseased hives noted by trained inspectors are brownish, clay-like, tenacious dissociations from infected degraded larvae and an uneven brood pattern [2,10]. At this late state of infection, the colony is no longer able to control infection by removing infected larvae. Dry-degraded larvae form a so-called putrefactive scale containing millions of infectious spores of *P. larvae* [2,10].

Therefore, the passive monitoring of symptoms of AFB in hives may not be efficient to protect colonies. Active surveillance is important for the early detection of infections at a subclinical stage in order to respond with appropriate measures to prevent disease outbreaks in honey bees. Therefore, the detecting of *P. larvae* spores in honey, in bees and in beehive waste at low concentrations is essential. At least in some countries, active monitoring is already carried out voluntarily [11]. 

The cultivation of *P. larvae* is based on spore germination and is therefore quite time-consuming. Although various molecular biological methods have been described, detection is mainly based on vegetative *P. larvae* or their residues in samples from beehives [12,13,14]. The isolation of DNA from *P. larvae* spores directly by lysing of the spore was not the focus so far, but may be relevant for the detection of non-germinating spores under laboratory conditions and for a timely detection [15]. 

In this study, we compared the results of two methods, the culture detection and RT-PCR detection of DNA directly isolated from spores. We analyzed honey samples in the framework of a voluntary screening program for five years in a district in the western part of Lower Austria, with the aim to detect subclinical infections in bee colonies. 

## 2. Materials and Methods

### 2.1. Beekeepers and Sampling under the Voluntary Monitoring Program

In the last five years, the samples outlined in Table 1 have been analyzed. The samples were either honey from honey chambers or from brood chambers. Honey was scraped out from the combs with a disposable teaspoon. Sampling was performed in spring between February and May (Table 1). All results were provided to the owners and reported to the local veterinary officer. If deemed necessary, the official veterinarian provided information to the beekeeper on how to reduce or eliminate spore load in bee colonies.

### 2.2. Cultural Detection of P. larvae

The culture-based method for detecting *P. larvae* was carried out according to the protocol of the Niedersaechsischen Landesinstituts für Bienenkunde, Herzogin-Eleonore-Allee 5, Celle [12]. The standard control spore-suspension batch no. RSK 16-07-21 was acquired from the same institute. Samples were diluted with an equal amount of sterile, distilled water and filtered through a sterile gauze swab (Covetrus, Brunn am Gebirge, Austria) to remove wax residues. Each sample (of 200 µL) was streaked out in triplicate onto Columbia-blood-agar plates with 5% sheep blood (Columbia agar plate with 5% Sheep blood, bio Mérieux, Vienna, Austria) using an L-Shaped Spreader Blue (VWR International, Vienna, Austria). Likewise, the spore standard was streaked out. After an incubation period of six days at 37 °C, the colonies were counted and verified using MALDI-TOF-MS identification (Maldi Biotyper^®^, Bruker Daltonics, Bremen, Germany). To calculate spores per gram of honey, the number of colonies was multiplied by 8.7 (based on the density of 1.35 g per mL of the spore solution, which roughly corresponds to the density of honey, and the volume of 200 µL diluted by 1:2).

### 2.3. DNA Extraction

The spores were de-coated according to the protocol of D’Alessandro et al. [15] with slight modifications: Ten grams of honey (or standard spore solution) were dissolved in 15 mL of buffer 1 (0.02 M Tris(hydroxymethyl)aminomethane-HCl, pH 8, 0.005 M ethylenediaminetetraacetic acid disodium salt dihydrate), supplemented with BashingBeads of Quick-DNA^TM^ Fecal/Soil Microbe Microprep Kit (Zymo Research, Irvine, CA, USA). After centrifugation at 20,000× *g* for 30 min, the pellet was suspended in 1.2 mL of buffer 2 (0.05 M Tris(hydroxymethyl)aminomethane, pH 10, 8 M urea, 1% *w*/*v* dodecyl sulfate sodium salt, 0.05 M ethylenediaminetetraacetic acid disodium salt dihydrate, 0.05 M dithiothreitol) and transferred to the BashingBead Lysis Tube. After incubation with shaking (600 rpm) at 60 °C for 1 h and centrifugation at 12,000× *g* for 20 min, the pellet was diluted in 400 µL of buffer 1 and 0.002 g/mL lysozyme and added with shaking for 40 min at 37 °C of 0.8 mg proteinase K and a further incubation with shaking for 40 min at 50 °C. After adding 750 µL of BashingBead buffer and bashing for 40 min (Retsch MM 200, Haan, Germany) at 30 Hz, the DNA was isolated according to manufacturer’s instructions (Quick-DNA^TM^ Fecal/Soil Microbe Microprep Kit, Zymo Research, Irvine, CA, USA). To investigate the isolation and detection of *P. larvae* spores by RT-PCR with and without de-coating for DNA isolation, selected samples with different spore colony counts were used. The DNA from these samples were isolated twice, with and without the de-coating procedure.

### 2.4. Real-Time-PCR (RT-PCR) 

*P. larvae* was detected using a commercial kit (BactoReal^®^ Kit American Foulbrood 1.1, Ingenetix Vienna, Austria) according to the manufacturer’s recommendation on a 7500 Fast Real-Time PCR System (ThermoFisher Scientific, Dreieich, Germany). A cycle threshold (CT) was calculated using internal software and a set of samples was taken and compared to the average colony counts of those samples.

The estimated logistic regression probability for various CT-score metrics was calculated for sensitivity and specificity [16].

A comparison of both methods, cultural and direct spore DNA isolation followed by RT-PCR, is shown in Figure 1.

### 2.5. Classification of Results

An average of all three colony plate counts was determined. A limit was set for “low spore counts” between one and twelve colony plate counts and “high spore counts” with thirteen and more colony plate counts. In addition, the apiaries were divided into three contamination classes: Class 0 for apiaries with samples in which no *P. larvae* could be detected, Class I for apiaries with at least one sample with low bacterial colony counts and Class II for apiaries with at least one sample of high bacterial colony counts [12].

### 2.6. Statistical Calculation of a CT-Value Using RT-PCR and Colony Counts and Sensitivity and Specificity

For fitting RT-PCR results to results from cultural microbiological analysis and to statistically calculate a suitable CT-value for RT-PCR, 80 field samples were analyzed both by cultural microbiology and RT-PCR. Results from cultural analysis were coded as 0/1 (i.e., no colonies/colonies detected) and CT-values grouped in domains (ranges). For each of these domains, the fraction of samples positive in cultural analysis was calculated. Logistic regression [16] was applied to obtain a statistically significant correlation (*p* < 0.05) of this fraction to the ranges of CT-values using the R statistical software. The number of samples and percentage of positive samples are listed in Table 2. 

Metrices for sensitivity and specificity were calculated based on the estimated probability of logistic regression for different CT-values, along with their 95% confidence intervals (calculates with R), and the cut-off CT was estimated.

### 2.7. Prevalence of P. larvae

For estimation of the prevalence of *P. larvae* in the bee populations in the study area in a given year, we related the number of beekeepers with at least one beehive infected with *P. larvae* to the total number of beekeepers, which had sent samples in the respective year. The 95% confidence interval was calculated according to Sachs [17] and a trend was calculated for the prevalence with MS-Excel (Excel 2019; Microsoft Corp., Redmond, WA, USA). Significance of the trend was assessed by *F*-test using MS-Excel.

## 3. Results

### 3.1. Monitoring

Clinical symptoms of AFB were observed in only one of the bee colonies in 2022. Samples were classified as “no spore counts”, “low spore counts” with an average colony count of between one to twelve colonies and “high spore counts” with an average of more than twelve colonies. In addition, apiaries were divided into three classes, as described in the Materials and Methods section.

The proportion of apiaries in class 0 was 92% in 2018 and 72% in 2019. Class II always included apiaries, ranging from 0% in 2019 to 5.6% in 2022. Samples with high bacterial colony counts were rarely identified, with the highest frequency in 2018, the first year of monitoring (5.2%).

The number of apiaries with at least one cultural positive sample per year was 5/62 (8%) in 2018, and it increased to 12/43 (28%), whereafter a linear decreasing trend was observed (Figure 2). The observed linear trend from 2019 to 2022 was statistically significant (*F* = 60.9; df = 2; *p* < 0.05).

Eleven beekeepers sent samples over all five years. Seven of them did not have any samples, in which spores were detected (Class 0). One beekeeper had samples with a low spore content over two years. Three beekeepers even had samples of high spore content in one year. 

In total, 793 samples were analyzed between 2018 and 2022 over the five-year voluntary program. Of these samples, 709 tested negative, thus no spore isolation was possible. A low spore content was seen in 67 samples and a high spore content was detected in 17 samples. The samples with positive detection over the five years are given in Table 3.

### 3.2. Sensitivity and Specificity of the RT-PCR Detection Method 

Metrices for sensitivity and specificity were calculated based on the estimated probability of logistic regression for different CT-values. In addition, a 95% confidence interval was determined for both for sensitivity and specificity indices (see Table 4). Based on a balanced sensitivity and specificity, a cut-off value CT ≤ 38 was estimated. 

In our study, we detected four samples that were cultural negative and showed a CT-value below 38. These samples originated from four different apiaries. Three of these beekeepers had also sent cultural positive samples in the same year.

### 3.3. Detection Limit with and without De-Coating

The CT-values of the RT-PCR of selected samples are similar for both methods, but for the spore standard RSK 16-07-21, the de-coating method results in a lower limit of detection (Table 5). Thus, the de-coating method is able to detect DNA directly from spores without the need for sporulation. This can save relevant time (Figure 1) compared to the cultural method. A typical RT-PCR plot of samples can be found in Appendix A. 

## 4. Discussion

AFB, caused by *P. larvae,* a spore-forming Gram-positive bacterium, is a serious disease of honey bees. Screening and surveillance are relevant to protect beekeepers’ colonies, in addition to certain hygienic measures that can be implemented. These measures are time-consuming and costly and should be carried out at an early stage of infection or even before *P. larvae* can be detected in regions with a high risk of infection. It is detrimental that clinical signs of the disease appear at a fairly late stage of the disease, often when the bee colony is already irretrievably lost. For early detection, PCR-based methods have an enormous time advantage—even more so when the detection method is based on DNA isolated directly from spores. The results of this study show that the use of a de-coating protocol to isolate DNA directly from spores of this pathogen has a timely advantage of more than five days. Especially in spring time when the bees’ breeding time starts, clinical outbreaks of AFB can be avoided by taking appropriate measures to reduce *P. larvae* [11]. The use of antibiotics is banned in bee colonies in Europe and New Zealand. Only vegetative cells of *P. larvae* can be attacked by antibiotics, but not spores, which are most important for dispersal, and therefore, the contamination of bee colonies and their products remain [18,19]. In our study, these measures were recommended for bee colonies with a high spore content (Class II apiaries). Beekeepers’ associations and official veterinarians that were informed about the results provided information about these measures which include the separation of adult bees from the brood pattern, beeswax, honey and pollen eradication and placing adult bees in a new, uncontaminated hive and feed sugar solution until they were able to feed themselves. All tools to be used must be free of spores, followed by an extensive inspection and monitoring once breeding has started. Locke et al. [11] have been successful in reducing AFB from 74 to 4% by monitoring bees in autumn followed by the application of appropriate measures in early spring. Their monitoring procedure can be compared with our cultural analyses of honey as it provides information over a longer period of time [20]. Nevertheless, in our study, a much lower percentage of bee colonies harbored spores in the beginning of the monitoring period (74% versus 8.1%). A constant reduction in apiaries with cultural positive samples from the second year to the fifth year was seen in our study as well (Figure 2), contrary to the first year with the lowest contamination rates. This might have been due to the sampling period (Table 1) or to the weather situation with rather low temperatures in January to March in the year 2018 with a late start in bee breeding (Appendix A, Austrian Weather Service). Various samples have been used to monitor and study *P. larvae*, e.g., honey, bees, hive debris or powdered sugar (applied on the upper bar and collected on a piece of paper on the base plate) [21]. The results on honey, long-living bees and beehive debris report over a period of time, while short-living summer bees and powdered sugar only represent a snapshot [19,21,22]. Honey from honey chambers or from brood chambers were used for our monitoring program. During our monitoring period, only one beekeeper kept two bee colonies showing clinical signs of AFB—these were eradicated. In general, the proportion of bee colonies without the detection of spores from *P. larvae* was high (2018/91.9%, 2019/72.09%, 2020/74.6%, 2021/81.35%, 2022/84.5%). Low spore counts were found in most *P. larvae*-positive bee colonies. In 2018, the proportion of beekeepers (4.83%) with a higher spore count was slightly higher (Class II). Otherwise, the spore loads were generally low throughout the monitoring period compared to other countries [11,23,24]. During the monitoring program, apiaries with detectable spores decreased from 2019 to 2022 from 28.91% to 15.5%. In the first year of monitoring, there were only 8.1% of apiaries with culturable spores in their samples; this might be due to rather cold weather compared to the following years and the sampling starting in early spring time.

Several methods have been described for PCR detection on vegetative *P. larvae* while not much has been reported on the direct isolation of DNA from spores [25]. Rusenova and co-workers showed, in a conventional PCR, that when using the de-coating buffer, it was possible to detect 10 to 46 colony-forming units per gram of sample volume with a sensitivity of 70% [26]. D’Alessandro and co-workers reported even higher sensitivity and specificity when using an adapted version of a de-coating buffer [15] in honey samples spiked with 0.1–10^6^ colony-forming units. Other authors were able to lower the limit of detection to one-to-eight spores per gram of honey using RT-PCR or TaqMan PCR [13,27].

Here, we adapted a method to directly isolate DNA from *P. larvae* spores to be used for RT-PCR. The imperative use of a de-coating buffer [15] for the extraction, followed by a commercial extraction kit with a bash-beating step to lyse spores, has been found to be most effective with a clean RSK 16-07-21 spore solution. A positive signal was detected with this method after just 31 PCR cycles instead of 40 cycles (without a de-coating buffer). However, similar results were observed in our field samples with and without the use of a de-coating buffer. These results could be due to an additional centrifugation step using the spore-de-coating protocol that segregates DNA from vegetative *P. larvae*. In order to reduce the risk of a clinical onset of AFB, a sensitivity of 100% and a specificity of 57% are proposed [28]. Reports of 29.5% of samples being bacteriologically negative but within the cut-off value [29] did not meet our standards.

Three of the four samples that detected positive in RT-PCR but negative for cultural spore isolation came from apiaries with additional cultural positive samples. 

The use of this fast detection method, delivering results after 6 h, could provide an effective tool to monitor AFB in spring time using honey samples and to take efficient control measures in time to protect bee colonies from AFB clinical outbreaks.

## 5. Conclusions

During a five-year voluntary monitoring, the fraction of apiaries with samples that were positive for *P. larvae* decreased from the second to the fifth year of monitoring. Although all 793 samples originated from bee colonies without any signs of *P. larvae* infection, the bacterium was detected in 11.8% of the samples. Thus, the early detection of such clinically inapparent, infected bee colonies is pivotal to establish corrective actions to limit intensity and the spread of the infection. We could show that the detection of *P. larvae* DNA in honey samples by RT-PCR without the need of a de-sporulation can yield results within 6 h. Such rapid detection is essential for the effective monitoring of AFB in spring time and allows us to take early and efficient control measures. 

## Figures and Tables

**Figure 1 vetsci-10-00213-f001:**
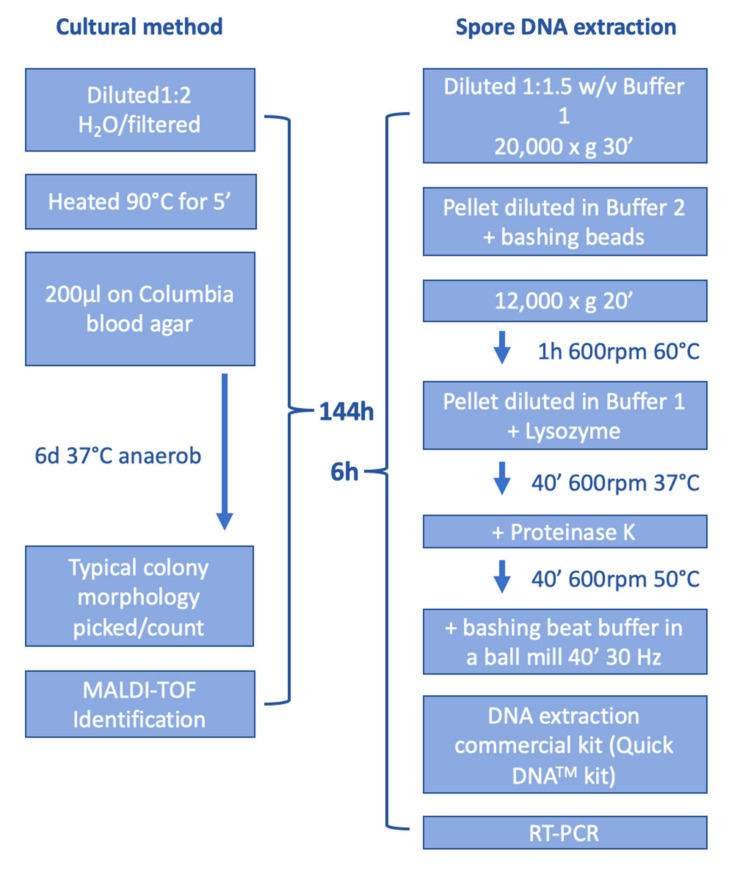
Detection of *P. larvae*. The cultural method takes 144 h whereas the direct DNA isolation from spores followed by RT-PCR takes 6 h.

**Figure 2 vetsci-10-00213-f002:**
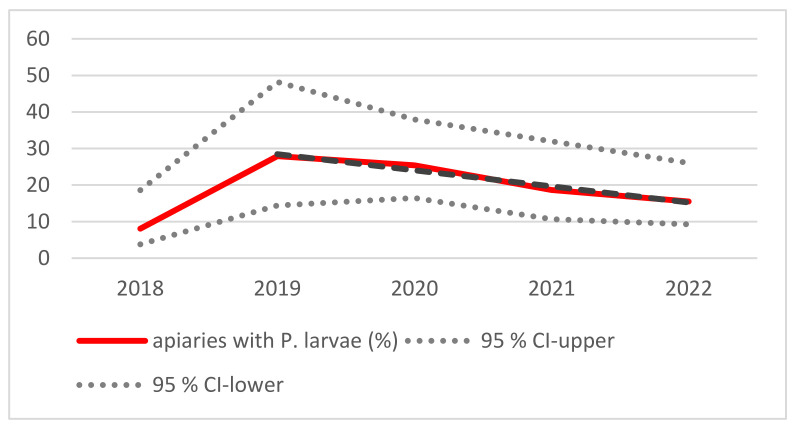
The percentage of apiaries with *P. larvae* detected (red line) and the 95% confidence interval (dotted grey lines). The dashed black line shows a linear declining trend for the years 2019–2022 (y = −4.4 * x + 32.9; R^2^ = 0.97, where x = duration in years, and y = the percentage of apiaries).

**Table 1 vetsci-10-00213-t001:** Samples from beekeepers in the years 2018–2022.

	2018	2019	2020	2021	2022
Sampling time	21.03–12.04 *	17.04–14.05	31.03–22.04	11.03–15.04	22.02–06.04
Samples	155	111	139	155	206
Beekeepers	62	43	63	59	71

* 21 of March to 12 of April.

**Table 2 vetsci-10-00213-t002:** Number of positive samples and CT-value categories.

CT-Value Range	Number of Samples	Culture Positive %
(23.0–31.2)	9	100
(31.2–33.9)	10	100
(33.9–35.9)	8	62.5
(35.9–37.9)	7	28.6
(37.9–38.6)	7	14.3
(38.6–43.2)	8	25.0
(45)	31	0.0

**Table 3 vetsci-10-00213-t003:** Number of samples over the years.

Year	Number of Samples		
	No Spores	Low Spore Content	High Spore Content
2018	143	4	8
2019	98	13	0
2020	116	20	3
2021	146	18	1
2022	206	12	5

**Table 4 vetsci-10-00213-t004:** Sensitivity and specificity of different CT-values following CT-value limits within the 95% confidence interval.

CT-Value	Sensitivity			Specificity		
	Estimated Value	Lower Limit	Upper Limit	Estimated Value	Lower Limit	Upper Limit
35	0.793	0.655	0.931	1.000	1.000	1.000
36	0.828	0.690	0.966	0.941	0.863	1.000
37	0.862	0.724	0.966	0.902	0.823	0.980
38	0.897	0.759	1.000	0.824	0.706	0.922
39	0.931	0.828	1.000	0.725	0.588	0.843

**Table 5 vetsci-10-00213-t005:** Sensitivity and specificity of different CT-values following CT-value limits within the 95% confidence interval.

Sample	Colony Count	CT-Value Using De-Coating	CT-Value without De-Coating
3795/1	10	31.2	29.8
3795/2	2	33.9	34.2
4/22	0	35.3	35.3
32/22	53	26.3	26.5
57/22	0	37.9	39.2
58/22	2	34.0	33.0
60/22	14	30.6	32.3
185/22	1	33.9	32.8
3796/2	2	33.8	32.1
RSK-16-07-21	26	31.0	40.0

## Data Availability

All data are availability on request.

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
