# Peer review of "Monitoring of Paenibacillus larvae in Lower Austria through DNA-Based Detection without De-Sporulation: 2018 to 2022"

_vetsci, 2023, doi:10.3390/vetsci10030213_

Round 1
Reviewer 1 Report
The article, "Monitoring of Paenibacillus larvae in Lower Austria from 2018 2 to 2022", focuses on the monitoring of P. larvae through a five-year voluntary initiative in western region of Lower Austria utilizing conventional pathogen detection and direct DNA-based detection methods without the need for desporulation. P. larvae is well-known for beekeepers as a severe pathogen causing the fatal honey bee disease American foulbrood, and other members of the genus are either secondary invaders of European foulbrood or considered a threat to honey bees. Hence, monitoring based on screening allows for the early detection of infections and the protection of bee colonies through the use of hygiene practices. Besides the novelty of the topic, the article has serious major concerns in addition to several grammatical and typographical errors, as mentioned below:
i) The article does not have a section on statistical analysis. In a similar vein, I failed to spot any instances of statistics use in the article, particularly in the tabular data. The results are not based on a statistical analysis of the data, which is completely worthless.
ii) The results section is quite vague and imprecise. I'm not sure if the study is about standardizing or optimizing the RT-PCR procedure or about the monitoring of P. larvae in honey bee spores; considering that the article focuses more on the sensitivity and specificity of CT values than it does on the fundamental idea of the study.
iii) Discussing section is very poorly written. The study's main findings are not discussed here. It's more like a general conversation regarding the subject. There is no justification for the findings, nor is there any comparison of the significant results with those from other investigations.
iv) Conclusion section is missing!!
The article has serious flaws in the sections of statistical analysis, results, discussion and conclusion. Before being taken into consideration, the article needs to be critically reviewed and revised.
Author Response
Dear reviewer 1
The article, "Monitoring of Paenibacillus larvae in Lower Austria from 2018 2 to 2022", focuses on the monitoring of P. larvae through a five-year voluntary initiative in western region of Lower Austria utilizing conventional pathogen detection and direct DNA-based detection methods without the need for desporulation. P. larvae is well-known for beekeepers as a severe pathogen causing the fatal honey bee disease American foulbrood, and other members of the genus are either secondary invaders of European foulbrood or considered a threat to honey bees. Hence, monitoring based on screening allows for the early detection of infections and the protection of bee colonies through the use of hygiene practices. Besides the novelty of the topic, the article has serious major concerns in addition to several grammatical and typographical errors, as mentioned below:
We changed according to reviewers statements, added a statistics for the screening data and highlighted the statistical part on the comparison of methods.
i) The article does not have a section on statistical analysis. In a similar vein, I failed to spot any instances of statistics use in the article, particularly in the tabular data. The results are not based on a statistical analysis of the data, which is completely worthless.
In the revised version, we explained more clearly the statistical methodology associated with the cut-off value estimation (2.6 Statistical calculation of a CT-value using RT-PCR and colony counts and sensitivity and specificity). To this end, some parts of the Results section were shifted to section 2.6.
We added a section (2.7) on prevalence estimation (apiaries with at least one cultural-positive sample/year), including methodology of trend analysis and significance testing.
ii) The results section is quite vague and imprecise. I'm not sure if the study is about standardizing or optimizing the RT-PCR procedure or about the monitoring of larvaein honey bee spores; considering that the article focuses more on the sensitivity and specificity of CT values than it does on the fundamental idea of the study.
We changed and rephrased the result section for clarity and importance. In particular, results of the voluntary monitoring are presented in more detail (Figure 2), and a declining trend in the fraction of apiaries with at least one cultural-positive sample/year for the years 2019-2022 is presented. We discussed the fact that in 2018 (the first year of the monitoring) the prevalence was low, and rose in 2019. A possible explanation for the low prevalence in 2018 is given (different weather situation in February 2018 compared to the following years) and supplementary meteorological information is contained in Figure S2.
iii) Discussing section is very poorly written. The study's main findings are not discussed here. It's more like a general conversation regarding the subject. There is no justification for the findings, nor is there any comparison of the significant results with those from other investigations.
The discussion was rewritten and comparison with other studies has been highlighted.
iv) Conclusion section is missing!!
We added a conclusion section.
The article has serious flaws in the sections of statistical analysis, results, discussion and conclusion. Before being taken into consideration, the article needs to be critically reviewed and revised.
We hope that the new version of the manuscript is more appropriate.
Reviewer 2 Report
This is an interesting paper on an important bee disease which tends to require drastic measures when detected too late. Improvement of detection is crucial and the authors present data about an "early detection method" that will provide information based on hive products (honey) where spores may be found before larvae are infected. This would be a very useful tool for beekeepers and the authors describe their procedures in reasonable detail (I made some additional suggestions see attached file). However, my main concern is that although the authors had access to so many samples, there is little information about the management decisions that derived from the early detection. Did the beekeepers use a "shook swarm" to save those colonies? How many were successful? Was there a seasonality trend to the positive samples. These epidemiological questions are important and it seems that you may have access to valuable data that could be added or presented in some way. This, in my opinion, would greatly support the value of a technique for early detection.

Author Response
Dear reviewer 2,
Many thanks for your valuable review!
Monitoring of Paenibacillus larvae in Lower Austria from 2018 to 2022
This is very important paper as the American foulbrood (being deadly disease that cannot be cured) is a persisting problem in beekeeping in many countries. I like the most important results very much and that is a fully described method for direct isolation of DNA from P. larvae spores without the need for desporulation. For that reason, this paper deserves to be published.
Dear reviewer, many thanks for your detailed and constructive review, the manuscript will highly profit from your comments.
The title has been changed accordingly.
However, prior to publication, manuscript has to be corrected, mainly because of improper organization (in ’Material and Methods’ and ’Results’ section). There are also some lingiustic errors and conceptual confusions (especially related to categorisation of beekeepers, sampling .... please see attached PDF where I marked and commented these confusions).
We have added or changed according to your valuable comments in the document.
The most important comments are:
TITLE:
The title should be improved as the main result is not mentioned. I suggest to expand the title as follows: „Monitoring of Paenibacillus larvae in Lower Austria from 2018 to 2022 with improved method for DNA based detection without the need for desporulation“
Changed
- MATERIAL AND METHODS:
- I would like to see the experimental design, simple and clear. It can be provided in a form of scheme (as a figure) or table.
We included a figure to outline both methods including a timeline.
- RESULTS:
- Subsection 3.1, Lines 127-129; Shouldn't it be "colonies" instead of beekeepers? Please consider this here and in subsection 3.5 and everywhere else in the text when you write about this.
Subsection 3,5: the whole subsection including Figure 4: Again, beekeepers or colonies? I am rather sure that you think of colonies.
We clarified this in the new methodology and results sections.
- Subsection 3.4, Lines 152-154: This belongs to the METHODOLOGY. In fact, it looks to me that the "isolation of DNA and detection of P. larvae WITHOUT de-coating of spores" is mentioned here for the first time. If I am right, please give the explaination in Material and Methods section (in subsections 2.3 or 2.4). Besides, methods WITH and WITHOUT de-coating of spores are not mentioned in the aim of the study (last paragraph of the Introduction).
We changed this according to your suggestions.
There are also other comments (in attached PDF) I inserted with the aim to help the authors to correct the manuscript.
Again many thanks for your valuable input, we took all your comments from your pdf into account.
Reviewer 3 Report
Monitoring of Paenibacillus larvae in Lower Austria from 2018 to 2022
This is very important paper as the American foulbrood (being deadly disease that cannot be cured) is a persisting problem in beekeeping in many countries. I like the most important results very much and that is a fully described method for direct isolation of DNA from P. larvae spores without the need for desporulation. For that reason, this paper deserves to be published.
However, prior to publication, manuscript has to be corrected, mainly because of improper organization (in ’Material and Methods’ and ’Results’ section). There are also some lingiustic errors and conceptual confusions (especially related to categorisation of beekeepers, sampling .... please see attached PDF where I marked and commented these confusions).
The most important comments are:
TITLE:
The title should be improved as the main result is not mentioned. I suggest to expand the title as follows: „Monitoring of Paenibacillus larvae in Lower Austria from 2018 to 2022 with improved method for DNA based detection without the need for desporulation“
2. MATERIAL AND METHODS:
- I would like to see the experimental design, simple and clear. It can be provided in a form of scheme (as a figure) or table.
3. RESULTS:
- Subsection 3.1, Lines 127-129; Shouldn't it be "colonies" instead of beekeepers? Please consider this here and in subsection 3.5 and everywhere else in the text when you write about this.
Subsection 3,5: the whole subsection including Figure 4: Again, beekeepers or colonies? I am rather sure that you think of colonies.
- Subsection 3.4, Lines 152-154: This belongs to the METHODOLOGY. In fact, it looks to me that the "isolation of DNA and detection of P. larvae WITHOUT de-coating of spores" is mentioned here for the first time. If I am right, please give the explaination in Material and Methods section (in subsections 2.3 or 2.4). Besides, methods WITH and WITHOUT de-coating of spores are not mentioned in the aim of the study (last paragraph of the Introduction).
There are also other comments (in attached PDF) I inserted with the aim to help the authors to correct the manuscript.

Author Response
Dear reviewer 3,
Many thanks for your comments and suggestions.
Monitoring of Paenibacillus larvae in Lower Austria from 2018 2 to 2022
Introduction
Line 45 – _I would suggest changing “cause” for “source”
This sentence was rephrased.
Line 58 – _form instead of from?
Thanks for spotting, changed.
Line 65 – _cites a study by Locke et al. 2019 – _in this study a 5-year monitoring program helped identify colonies that had low level infections in the fall and could be treated in the Spring. The program showed a marked reduction in infection over time.
I am curious about how the result of this study compares to the results shown in Locke et al. How may timing of the screening procedure affect the success for this method in colony management? Were all the samples taken around the same time of the year?
I am also curious of turn-around time in the result delivery and recommended management options after detection of low spore count (the authors indicated that high spore count colonies were destroyed).
We added a paragraph in the discussion section.
If the standard procedure based on spore germination is lengthier (probably should indicate how long it usually takes) and the proposed screening is faster (time frame for results) and if this screening allows you to detect infections earlier by concentrating in sampling honey, maybe a graphic illustrating the similarities and the differences (in procedure -including ease of collection and mailing of samples, time for in-house testing and result delivery) could be very useful for the reader.
We added a new figure in the revised manuscript.
This is not my field of expertise but creating a Ct curve (including calibration and threshold levels) is tricky so I won’t comment on the details of the procedure, however, It might be good to include a graph with the amplification plot and the threshold range.
We added a supplementary Figure S1 in the revised manuscript.
Results
Lines 238 -244
I think it is very interesting that you were able to detect numerous low-level infections. In the years 2019 and 2020 there around 25 % of samples with detectable, albeit low spore counts.
My feeling is that you have 2 important things to report here
1-your technique and the accuracy of early detection
2- the epidemiology part of the apiaries that were sampled …and this part I think it is highly important and not extensively developed in your paper.
We reconstructed the results and discussion section to highlight the epidemiological part.
What happened to those colonies that very positive for low spore counts (1/4 of all the samples in those 2 years) The official authority informed beekeepers about the results and effective hygiene measures to avoid disease development. None of these beekeepers reported a clinical disease outbreak. We mentioned this in the new results section and discussed it.
How were they treated?
See above and in the modified manuscript.
Did early detection prevent development of the disease?
See above and in the modified manuscript.
I think that you have important data that you should attempt to organize and present. The value of a new detection technique for American foulbrood gains more “power” when put into practical context.
Many thanks for your valuable comments, we reorganized and presented the data in a more appropriate way in the new version.
Round 2
Reviewer 1 Report
I appreciate the authors' efforts to fix my concerns and improve the manuscript.
I have a few minor comments:
1. The title could be modified as “Monitoring of Paenibacillus larvae in Lower Austria through DNA-based detection without desporulation: 2018 to 2022”.
2. Line No. 35-36 & in tabular data at Line No. 155-156: I’m concerned the authors have mistakenly used a comma (,) between values instead of full stop (.). Revise where applicable.
3. Line No. 39: Rearrange the keywords as: American Foulbrood; Paenibacillus larvae; Spore; RT-PCR
4. Line No. 168 (section= Results): Paragraphs in this section are too small; some even contain just one sentence.
5. Line No. 298 (section= conclusion): This chapter must contain the authors' conclusive remarks based on the current study's findings, which are not reflected here. I suggest modifying this section and avoiding using irrelevant wordings/ sentences.
Remarks: There are still a lot of typographical and grammatical errors in the manuscript, which need to be fixed before the possible publication of the article.
Author Response
Review 2
I appreciate the authors' efforts to fix my concerns and improve the manuscript.
Dear reviewer, thank you vey much for this comment. We have now carefully revised the manuscript with respect to the comments listed below.
I have a few minor comments:
- The title could be modified as “Monitoring of Paenibacillus larvaein Lower Austriathrough DNA-based detection without desporulation: 2018 to 2022”.
The title has been reworded accordingly.
- Line No. 35-36 & in tabular data at Line No. 155-156: I’m concerned the authors have mistakenly used a comma (,) between values instead of full stop (.). Revise where applicable.
We have checked all numerical expressions and now, full stop and comma are used correctly.
- Line No. 39: Rearrange the keywords as: American Foulbrood; Paenibacillus larvae; Spore;RT-PCR
Keywords have been rearranged
- Line No. 168 (section= Results): Paragraphs in this section are too small; some even contain just one sentence.
We arranged sentences to form longer paragraphs.
- Line No. 298 (section= conclusion): This chapter must contain the authors' conclusive remarks based on the current study's findings, which are not reflected here. I suggest modifying this section and avoiding using irrelevant wordings/ sentences.
We reworded the conclusions section to have the conclusions based on our results.
Remarks: There are still a lot of typographical and grammatical errors in the manuscript, which need to be fixed before the possible publication of the article.
The manuscript has been revised for typos, grammar and syntax.